# A Simple Methodology to Estimate the Diffusion Coefficient in Pervaporation-Based Purification Experiments

**DOI:** 10.3390/polym11020343

**Published:** 2019-02-15

**Authors:** Gabriela Dudek, Przemysław Borys

**Affiliations:** Department of Physical Chemistry and Technology of Polymers, Faculty of Chemistry, Silesian University of Technology, Strzody 9, 44-100 Gliwice, Poland; p.borys@polsl.pl

**Keywords:** diffusion coefficient, pervaporation, ethanol/water separation

## Abstract

A procedure to estimate the diffusion coefficient in solution–diffusion models of hydrophilic membranes used in pervaporation-based purification experiments is presented. The model is based on a series solution of the general permeation problem. It considers a membrane that can be filled with water or with the feed solution before the measurement. Furthermore, the length of the tubing between the permeation cell and the place of cold traps is also addressed. To illustrate the parameter estimation procedure, we have chosen the data for the separation of water and ethanol by chitosan membranes. It is shown that the diffusion coefficient can be estimated effectively from the time course of the transported mass and by the analysis of certain well defined time lags of the permeation curve.

## 1. Introduction

Pervaporation (PV) is a promising technique employed widely in liquid separation such as removal of water from organic solvents, the purification of aqueous streams, and the separation of organic–organic mixtures [1,2,3,4,5]. Apart from being environment friendly, pervaporation offers numerous benefits such as low energy consumption, high selectivity, excellent separation ratio, modular and compact design. In pervaporation, the separation process occurs by the solution-diffusion mechanism, proposed by Graham, which takes place in three steps: sorption of liquid at the upstream side of the membrane surface; diffusion of mass through the membrane; and desorption of the sorbed molecules in vapour form at downstream side of the membrane under the action of the reduced pressure [6]. The solution-diffusion suggests that the selectivity and permeation rate are governed by sorption and diffusion of each component of the feed mixture to be separated. The desorption is considered to have no major influence on the pervaporation process due to maintaining a state close to the vacuum on the side of permeate—the partition between nearly zero concentration behind the membrane and nearly zero concentration in the membrane (but less than behind the membrane due to partition) is not relevant to the modelling. Sorption is a thermodynamic property and diffusion is a kinetic property and they constitute two different factors that affect transport. The sorption process is determined primarily by the chemical nature of the membrane polymer and permeating molecules, while the diffusion is determined by the physical structure of the polymer and the size of the permeating molecules [7,8,9,10].

Many attempts have been made to derive relationships to calculate the diffusion coefficients of molecules penetrating nonporous membranes in a pervaporation process. Yeom et al. [11] proposed a practical method to determine the diffusion coefficients from steady state pervaporation experiments of single components. The method relies on fitting a numerical solution of the diffusion equation to the experimental pervaporation data. The main principle of this method is based on Fujita’s free volume theory and Flory–Huggins thermodynamics. Huang et al. [12] developed the theoretical model for calculating the diffusion coefficients of liquids in polymer membranes. This model is also based on Fujita’s free volume theory and considers crystallinity in the polymer and was tested by the literature data, i.e., n-hexane-polyethylene and benzene–polyethylene systems. Walcher et al. [13] proposed the method to determine the diffusion coefficient as the two-parameter function $D=D_{0}\cdot e^{\gamma c}$ , i.e., the intrinsic diffusion coefficient *D*_0_ and the plasticisation parameter *γ*. The value of *D*_0_ was determined by fitting the ideal diffusion theory asymptotic solution to the experimental permeation data during the early part of the transient. The parameter *γ* was determined only in the stationary state of pervaporation.

Borys and Grzywna [14] analysed the long-time asymptotics of the solution for an initially evacuated membrane with an absorbing boundary condition on the right-hand side. Such long-time asymptotics, after certain manipulation of equations, gave an opportunity to calculate the basic transport parameters: the partition and diffusion coefficient.

A Maxwell–Stefan approach to model the transport in pervaporation is also frequently called for. However, this type of modelling rather applies to porous membranes, and application to nonporous membranes (where a solution-diffusion mechanism is expected) is not entirely along the line of this method [15,16].

Poly(vinyl alcohol) (PVA) [17,18], chitosan [19,20], alginate [21,22], polyimide [23,24] and polyaniline [25,26] have been employed as membrane materials for the dehydration of organic solvents, due to their good chemical stability, outstanding physicochemical properties and/or excellent membrane forming ability. However, hydrophilic membranes have to be stored in distilled water or the feed solution. Otherwise, the membrane undergoes corrugation and it is not possible to put the membrane correctly into the permeation cell. With this in mind, we proposed new method to estimate the diffusion coefficient, based on a series solution of the general permeation problem, where the analysed membranes are not required to be empty at initial stage of the measurement (which, besides the corrugation problem, would imply a problem with the modelling of membrane swelling). We also consider the impact of the tubing length between the permeation cell and the cold traps which causes a delay in the mass flow, in the first moments of measurement. As an illustration, computational analysis was addressed to the experimental results obtained for chitosan membranes cross-linked with glutaraldehyde.

In spite of the various, complex methods for determining the diffusion coefficient, the researchers usually do not evaluate the value of diffusion coefficient but only estimate parameters that describe the effectiveness of investigated membranes (e.g., *PSI*, flux and separation factor) [17,18,23,24,25,26]. Much less frequently sorption and diffusion coefficients are considered [8,27,28,29], which requires additional experimental setups. Such approach is in contrast to what we can find in the analysis of permeation of gases where determination of *D* is crucial. As a consequence in PV we typically are not able to determine which process—diffusion or solubility—is more responsible for the permeation of vapours through investigated membranes.

For this reason, it seems necessary to present a simple methodology that can be easily incorporated into experimental practice, if possible—without additional experimental approach. To keep things simple, among the others, we use a constant diffusion coefficient, and we ignore the influence of one component (water), on the transport of the other (ethanol)—we consider the other component to be just a part of the environment in which diffusion takes place. We also do not introduce the concentration-dependent diffusion coefficients for single components and use an effective diffusion coefficient $\overset{´}{D}$, related approximately to the functional concentration-dependent diffusion coefficient *D(c)* by the relation $\overset{´}{D}=\frac{1}{C_{0}}\int_{0}^{C_{0}}D\left(c\right)dc$ [30]. Such an approach, even though simple, still allows for dissection of the main transport effects—the diffusion and sorption. The entire methodology can be represented in the figures, as certain straight lines intersecting at particular points, the horizontal and vertical axes of the permeation curves at particular points and the shape of these curves can be understood without advanced preliminaries; it does not require any numerical simulations or advanced mathematical treatment.

In the presented approach we utilize the rarely cited analytical solutions for time lag with various initial solvent concentration profiles within the membrane. Despite the fact that such solutions have previously been published, no one has commented on transient shape of the permeation possible curves in pervaporation process, and there is no information about the correction of the recorded curves due to the influence of the tubing length.

## 2. Theoretical Background

The solutions to various transport problems in field of diffusion are presented in the classical book by Crank [30]. In chapter 4 the author considered various cases of the one-dimensional diffusion in a medium bounded by two parallel planes, e.g., the planes at *x =* 0, *x = l*. The total amount of the diffusing substance *Q_t_* which passed through the membrane after time *t* in a permeation problem,
(1)$$\frac{\partial c}{\partial t}=D\frac{\partial^{2}c}{\partial x^{2}}$$
for $c\left(x,0\right)=c_{0},c\left(0,t\right)=c_{1},c\left(l,t\right)=c_{2}$ can be calculated from the equation 4.24 of Crank [30]:(2)$$Q_{t}=D\left(C_{1}-C_{2}\right)\frac{t}{l}+\frac{2l}{\pi^{2}}\sum_{n=1}^{\infty }\frac{C_{1}cosn\pi -C_{2}}{n^{2}}\left\{1-exp\left(-Dn^{2}\pi^{2}t/l^{2}\right)\right\}\;\frac{+4C_{0}l}{\pi^{2}}\sum_{m=0}^{\infty }\frac{1}{{\left(2m+1\right)}^{2}}\left\{1-exp\left(-D{\left(2m+1\right)}^{2}\pi^{2}t/l^{2}\right)\right\}$$
where, *Q_t_*—total amount of diffusing substance per unit surface of the membrane; *C*_0_—the initial concentration in the membrane; *C*_1_—the concentration on the feed side of the membrane; *C*_2_—the concentration on the permeate side of the membrane; *t*—the time of pervaporation experiment; *l*—the thickness of membrane; and *D*—the diffusion coefficient. The units should be chosen self-consistent, e.g., *Q_t_* in mol/m^2^, *C*_0_, *C*_1_, *C*_2_ in mol/m^3^, *t* in *s*, *l* in *m* and *D* in m^2^/s. The amount of substance can also be measured in other units than mols, for example in kilograms, and the length unit can be replaced, for example, by cm (in *Q_t_*, *C*_0_, *C*_1_, *C*_2_, *l* and *D*).

In the most common experimental permeation arrangement, it is assumed that the initial concentration in the membrane and at the permeation side is equal to zero. In this case *C*_0_ = *C*_2_ = 0, and the equation 2 takes the form
(3)$$Q_{t}=DC_{1}\frac{t}{l}+\frac{2l}{\pi^{2}}\sum_{n=1}^{\infty }\frac{C_{1}cosn\pi }{n^{2}}\left\{1-exp\left(-Dn^{2}\pi^{2}t/l^{2}\right)\right\}$$

The example graph of the total amount of diffusing substance versus time, based on the Equation (3), for *D* = 1, *l* = 1 and *C*_1_ = 1 is shown in Figure 1.

When the time tends to infinity (t→∞), the function $exp\left(-Dn^{2}\pi^{2}t/l^{2}\right)$ tends to zero. The asymptotic form of Equation (3) can be written as a linear function of time:(4)$$Q_{t}=\frac{DC_{1}}{l}\left(t-\frac{l^{2}}{6D}\right)$$

An intercept *L_a_* on the t-axis takes the form
(5)$$L_{a}=\frac{l^{2}}{6D}$$

Hence, the diffusion coefficient can be calculated from the equation
(6)$$D=\frac{l^{2}}{6L_{a}}$$

This is the standard time lag equation, which (despite opinions like Jyoti et al [10], Ball et al [31], Yamaguchi et al [32]) is however not directly applicable to the considered pervaporation experiments, where $C_{0}\neq 0$ and where the tubing perturbs the recording of permeation curves. In our measurements for example, we obtain negative time lags which world predict negative diffusion coefficients. We extend the theory in the “Results and Discussion” section to address the real experimental setup.

The Equation (3) allows us to calculate the feed concentration in the membrane (and thus, knowing the concentration outside of the membrane, to evaluate the partition coefficient). Establishing *t* = 0 we have
(7)$$C_{1}=\frac{-6Q\left(0\right)}{l}$$

*C*_1_ can also be determined from a sorption experiment. A typical setup includes a container filled with liquid and covered by a membrane. The weight loss of the liquid with respect to time is monitored to calculate the sorption rate. This method has two basic disadvantages: it is not very accurate and such method is suitable for measuring concentration in the case when the membrane is immersed in a pure component. If the membrane is diluted in a mixture (e.g., water–ethanol), the proportion of components in the membrane will not be known [33]. To fix the later issue, there is another method that is based on the desorption of the previously sorbed liquid to vacuum in order to determine its composition by gas chromatography or other technique. Such a method has also some disadvantages. The total desorption of the liquid sorbed into the membrane cannot be guaranteed [29]. Additionally, large membrane samples that are required to collect sufficient liquid volume lead to component evaporation and accuracy loss, especially when working with very volatile substances.

## 3. Experimental

### 3.1. Materials

Chitosan (*MW* = 600,000–800,000 Da), acetic acid (purity ≥ 99.8%), glutaraldehyde solution (25% in H_2_O) and sodium hydroxide (purity ≥ 98%) were purchased from Acros Organics (Geel, Belgium).

### 3.2. Membrane Preparation

A 3% chitosan solution was prepared by dissolving an appropriate amount of chitosan powder (*MW* = 600,000–800,000 Da) in 1% aqueous acetic acid solution. The chitosan solution was then cast onto a levelled glass plate and evaporated to dryness at 40 °C. Next, after 24 h, the membrane was cross-linked using 1.25 wt % glutaraldehyde solution in water. Cross-linked glutaraldehyde membranes were prepared by 5 min washing of the dry membrane with 50 cm^3^ glutaraldehyde solution and subsequent washing with deionized water, 2 wt % sodium hydroxide solution and again deionized water until neutral pH was obtained. The membrane thickness was measured using waterproof precise coating thickness gauge MG-401 ELMETRON (Elmetron, Zabrze, Poland), estimated as a mean values of at least 10 measurements in different points and equal to 40.0 ± 2.0 μm.

### 3.3. Pervaporation Experiments

Pervaporation experiments were performed at room temperature using a pervaporation setup shown in Figure 2. The feed (1 dm^3^ of 96 vol % ethanol solution) was poured into the feed tank (1). The solution was supplied through a circulating pump (2) with velocity 9.25 × 10^−2^ m^3^h^−1^ on the high pressure top side of the membrane placed inside the permeation cell (3) assembled from two half-cells made of stainless steel and fastened with bolted clamps. The effective surface area of the membrane is 112 × 10^−4^ m^2^. The membrane was placed on a finely porous stainless steel plate support. Before measurement the circulating pump was turned on for 10 min without running the vacuum pump on the permeation side to check the system for leaks. This is the stage which inevitably results in a sorption of the feed solution in the membrane (and it gives a fingerprint in the mass transport curve, as will be shown later). After the ethanol/water separation in the membrane module, the retentate was recirculated back to the feed tank (1) while the permeate was received and frozen in the cold traps condenser unit (5). Due to technical limitations, the cold traps were not placed directly under permeation cell but there was a distance of about two meters between them. The reduced pressure on the permeate side equal to 300 Pa was produced by a Agilent SH110 vacuum pump (6) and controlled with a Pfeiffer TPG-201 vacuum gauge (4). The total amounts of diffusing substances were calculated from the measured weight of a liquid collected in the cold traps, using the analytical balance, during certain time intervals, i.e., initially every 1 min (during the first 10 min) and then every 5 min (during the next 110 min). The measurement was repeated twenty times.

## 4. Results and Discussion

### 4.1. Diffusion Coefficient Estimation Strategy

Hydrophilic membranes are stored in water or feed solution before experiment. It is also necessary to also consider that before the measurement the internal circulation has been activated causing the sorption of feed molecules into the membrane. It implies that the initial concentration C_0_ in the membrane, when turning on the vacuum pump, is not zero. In such case Equation (6) to calculate the diffusion coefficient does not apply. One must take into account the nonzero value of the C_0_ and solve Equation (1) to find a new equation for the diffusion coefficient.

The diffusion coefficient for solvents in polymer membranes can be strongly concentration-dependent; however, in case of a permeation process it is possible to use the average diffusion coefficient. This is not typically a simple average of the minimum and maximum diffusion coefficient present in the sample. One of the known estimates of such average, addressed to permeation experiments was proposed by Crank in ref. [30] by considering the stationary flux of mass in a permeation experiment, also analysed by Borys in [34]:(8)$$D=\frac{1}{C_{0}-C_{2}}\int_{C_{0}}^{C_{2}}D\left(c\right)dC$$

Using such averages for representing *D*, we are able to dissect the transport and solubility issues, while the theory remains simple.

#### 4.1.1. Diffusion Coefficient for the Membrane Filled with Feed Solution

When the membrane is filled with feed solution the initial concentration in the membrane is the same as the concentration on the feed side of the membrane (*C*_0_ = *C*_1_). The concentration on the permeate side is equal to zero. In this case the Equation (2) takes the form
(9)$$Q_{t}=C_{0}[\frac{Dt}{l}+\frac{2l}{\pi^{2}}\sum_{n=1}^{\infty }\frac{cosn\pi }{n^{2}}\left\{1-exp\left(-Dn^{2}\pi^{2}t/l^{2}\right)\right\}\frac{+4l}{\pi^{2}}\sum_{m=0}^{\infty }\frac{1}{{\left(2m+1\right)}^{2}}\left\{1-exp\left(-D{\left(2m+1\right)}^{2}\pi^{2}t/l^{2}\right)\right\}]$$

The example graph of the total amount of diffusing substance versus time, based on the Equation (9) for *D* = 1, *l* = 1 and *C*_1_ = 1, is shown in Figure 3. One can see a difference compared to Figure 1, namely the flux does not grow slowly from zero, but starts with a large nonzero value. This is a fingerprint of this situation, and is related to a rapid desorption of the permeate on the permeation side of the membrane.

When the time tends to infinity the exponentials in equation (9) vanish, reducing it to
(10)$$Q_{t}=C_{0}\left[\frac{Dt}{l}+\frac{2l}{\pi^{2}}\sum_{n=1}^{\infty }\frac{cosn\pi }{n^{2}}+\frac{4l}{\pi^{2}}\sum_{m=0}^{\infty }\frac{1}{{\left(2m+1\right)}^{2}}\right]$$

Hence, the asymptotic total amount of diffusing substance takes the form
(11)$$Q_{tas}=C_{0}\left[\frac{Dt}{l}+\frac{2l}{\pi^{2}}\cdot 1.644929\right]=C_{0}\left[\frac{Dt}{l}+\frac{l}{3}\right]$$
and the diffusion coefficient can be calculated using the following equation.
(12)$$D=\frac{-l^{2}}{3L_{a}}$$
(Note the minus, as the time lag is negative in this configuration.)

For *t* = 0 one can find *C*_0_ as $C_{0}=\frac{3\cdot Q_{tas}\left(0\right)}{l}$.

These equations are only valid if the additional time lag introduced by the tubing, that connects the membrane to the freezing chamber is negligible, i.e., the mass curve *Q(t)* should grow instantly at *t* = 0, and not at *t* = *t*_0_ (*L*_1_), after the mass diffuses through the tubing. The later case shifts the intersection of the asymptote with the axes of coordinate system and requires further analysis, which we present in the following paragraphs.

#### 4.1.2. Diffusion Coefficient for the Membrane Initially Filled with Water Only

When the membrane is filled with water (not feed solution, i.e., no other species of the binary mixture is there) the concentration in the membrane is not the same as the concentration on the feed side of the membrane (*C*_0_ ≠ *C*_1_). The concentration on the permeate side is equal to zero. In this case Equation (2) takes the form
(13)$$Q_{t}=\frac{DC_{1}t}{l}+\frac{2l}{\pi^{2}}\sum_{n=1}^{\infty }\frac{C_{1}cosn\pi }{n^{2}}\left\{1-exp\left(-Dn^{2}\pi^{2}t/l^{2}\right)\right\}\frac{+4C_{0}l}{\pi^{2}}\sum_{m=0}^{\infty }\frac{1}{{\left(2m+1\right)}^{2}}\left\{1-exp\left(-D{\left(2m+1\right)}^{2}\pi^{2}t/l^{2}\right)\right\}$$

The example graph of the water and ethanol diffusing molecules versus time, based on Equation (13), for 96 vol% of ethanol as a feed solution, i.e., *C*_1_ = 0.04, *C*_0_ = 1 and *C*_1_ = 0.96, *C*_0_ = 0, for water and ethanol, respectively, is shown in Figure 4.

One can see that the water curve displays a large nonzero initial flux (because water was present in the membrane at *t* = 0), while the ethanol curve displays no such behaviour. When the time tends to infinity, Equation (13) for water reduces to
(14)$$Q_{t}=\frac{DC_{1}t}{l}+\frac{2lC_{1}}{\pi^{2}}\sum_{n=1}^{\infty }\frac{cosn\pi }{n^{2}}+\frac{4C_{0}l}{\pi^{2}}\sum_{m=0}^{\infty }\frac{1}{{\left(2m+1\right)}^{2}}$$

Hence, the asymptotic equation for the total amount of diffusing water takes the form
(15)$$Q_{tas}=C_{1}\left[\frac{Dt}{l}+\frac{l}{6}\right]+\frac{C_{o}l}{2}$$
and the diffusion coefficient can be calculated using the following equation.
(16)$$D=\frac{l^{2}}{L_{a}}\left(\frac{1}{6}-\frac{C_{0}}{2C_{1}}\right)$$
where $L_{a}$ is the point where the asymptote crosses the time axis.

Taking in a first approximation a constant partition coefficient *K*, where
(17)$$K=\frac{C_{1}}{C_{feed}}$$
it is possible to write *C*_0_ and *C*_1_ (the case of 96 vol% solution of ethanol) for water as
(18)$$C_{0}=K\times 1.00\;\mathrm{and}\;C_{1}=K\times 0.04$$
(That is, at lower volume fraction the water molecules “attack” the surface of the membrane at lower rate.) So, finally (for a 96 vol% ethanol solution), one obtains
(19)$$D=-12,33\frac{l^{2}}{L_{a}}$$
(This equation should be individually calculated for differing concentrations of the feed solution, based upon Equation (16).)

#### 4.1.3. Considering the Effect of Tubing between the Membrane and the Cold Trap

The connection of the membrane and a cold trap is done by means of an additional tubing. This tubing does not establish an instantaneous transport and introduces additional delays compared to the time lags expected in Section 4.1.1 and Section 4.1.2.

The delay can be measured by the intersection between the tangent to the inflection point for curves (8) or (12) and the time axis. We call this time lag $L_{a}^{1}$ and it closely approximates the classical permeation time lag of the tubing (the permeation lag was introduced in Section 2). Knowing this time lag we must rescale it into the shift of the final mass asymptote and then we must decrease the asymptotic time lag by the calculated value to obtain an estimate of the membrane permeation time lag that is not influenced by the tubing delay.

The shift in the asymptote, as turns out from our numerical simulations for various tubing lengths and diffusion coefficients (Figure 5), is related to the time lag $L_{a}^{1}$ by
(20)$$\Delta t_{asymptote}=6.5\cdot L_{a}^{1}$$

The simulation was evaluated by random walk models as described e.g., in [35,36], where variations in the time step vs. lattice step reflect variations in diffusion coefficients.

In this view, the experimental configuration considered in Section 4.1.2 causes problems in the estimation of the time lag for the other species (B, ethanol in our example), which is initially at zero concentration. The other species does not display any inflection point, and we have no marks on the permeation curve that could allow the identification of the asymptote shift due to the tubing.

However, we can estimate the $L_{a}^{1}{}_{EtOH}$ based upon the $L_{a}^{1}{}_{H_{2}O}$ by the results from kinetic theory of gases, where the diffusion coefficient in binary mixture equals [33]
(21)$$D=\frac{3kT}{8pd_{AB}^{2}}\sqrt{\frac{RT}{2\pi }\left(\frac{1}{M_{A}}+\frac{1}{M_{B}}\right)}$$
where *k* is the Boltzmann constant, *R* is the gas constant, *T* is the temperature, *p* is the pressure, *d_AB_* = 0.5(*d_A_* + *d_B_*) is the average kinetic diameter of the diffusing molecules and *M_A_* and *M_B_* are the molar masses of species *A* and *B*, respectively, expressed in kilograms.

Because the diffusion coefficients are related to time lags ($D=l^{2}/\left(6\cdot L_{a}^{1}\right)$), we can use this equation to estimate the time lag for species B given time lag (or diffusion coefficient) for species A:(22)$$\frac{La_{A}^{1}}{La_{B}^{1}}=\frac{D_{Btub}}{D_{Atub}}=\frac{d_{AAir}^{2}}{d_{BAir}^{2}}\sqrt{\frac{\frac{1}{M_{B}}+\frac{1}{M_{Air}}}{\frac{1}{M_{A}}+\frac{1}{M_{Air}}}}$$

We may also estimate the tubing time lag for species B based only upon the Equation (19) and the relation $L_{a}^{1}=l^{2}/\left(6\cdot D\right)$—which of these two approaches to use is a matter of choice for the experimentalist. However, estimation of the lag for B based upon the lag for A (Equation (22)), given a not perfectly ideal setup for permeation leaves a chance to calculate both lags in similar experimental conditions and make them comparable. Knowing the time lag of the tubing we must rescale it into the shift of the asymptote and then we must decrease the asymptotic time lag by the calculated value to obtain an estimate of the permeation time lag that is not influenced by the tubing delay.

#### 4.1.4. Experimental Evaluation of the Shift in the Asymptote Due to Tubing

Instead of a sophisticated theoretical analysis, it is possible to address the problem of the perturbation introduced by tubing experimentally. Namely, one must perform a series of permeation measurements (i.e., three measurements) for different tubing lengths. The time lag $L_{a}^{1}$ (and the shift of the asymptote $L_{a}^{2}$ as well) scales with square of *L*. Therefore, knowing three such lags, we can determine this parabolic relation and extrapolate the asymptotic time lag $L_{a}^{2}$ to the case of *L* = 0, i.e., to the condition that applies to the ideal formulation of the problem and allows to use Equations (6) and (12).

#### 4.1.5. Application to Chitosan Pervaporation Membrane

The diffusion coefficient estimation procedure is applied to chitosan membranes investigated in ethanol/water separations. Figure 6 shows the average of twenty measurements a relation between the total amount of diffusing water or ethanol molecules and time.

It can be seen that initially the total amount of diffusing ethanol and water is nearly zero. It is due to the tubing that connects the permeation cell and the place of cold traps, where the permeate is collected. The permeate needs $L_{a}^{1}$ to arrive to the cold trap. The next segment is the region of the function’s greatest growth. From this part of the function it is possible to calculate time lag for the tubing $L_{a}^{1}$ (related to the inflection point). The asymptote of the mass transport function is linear. When the membrane is filled with permeating water and ethanol molecules the same amount of diffusing substances leaves membranes in the same period of time. From this part of the function shown in Figure 6 it is possible to evaluate the value of the asymptotic Time Lag $L_{a}^{2}$ (related to Equation (12)). But this time lag is in fact shifted by the delay introduced by the tubing (*L*_1_). This delay transforms to a delay of the asymptote by $\Delta t_{asymptote}=6.5\cdot L_{a}^{1}$.

The time lag for the tubing and asymptotic time lag for water are 1.42 and −10.55 min, respectively. $L_{a}^{1}$ and $L_{a}^{2}$ for ethanol are equal to 2.50 and −0.44 min, respectively. Hence, the effective total time lags, corrected by Equation (20), are $L_{aH_{2}O}=-19.78$ min and $L_{aEtOH}=-16.69$ min.

Additionally, the time lags $L_{a}^{1}$ should approximately relate to the diffusion coefficients of water and ethanol in the tubing. We can try to calculate them and compare to the predictions of kinetic theory by Chapman and Cowling [37] (Equation (21)). Having ${L_{a}^{1}}_{H_{2}O}$ = 1.42 min = 85 s and ${L_{a}^{1}}_{EtOH}$ = 2.5 min = 150 s, and taking tubing length equal to 2m from our data, evaluating the time lags: $D_{H_{2}O}=\frac{4m^{2}}{6\cdot 85s}=7.8\cdot 10^{-3}m^{2}s^{-1}$ and $D_{EtOH}=\frac{4m^{2}}{6\cdot 150s}=4.4\cdot 10^{-3}m^{2}s^{-1}$.

The kinetic theory for $M_{H_{2}O}=0.018kg\cdot mol^{-1}$, $M_{EtOH}=0.046kg\cdot mol^{-1}$, average $M_{Air}=0.0289kg\cdot mol^{-1}$ and kinetic diameters $d_{H_{2}O}=2.68\overset{´}{Å}$, $d_{EtOH}=4.50\overset{´}{Å}$, average $d_{Air}=3.60\overset{´}{Å}$ results in $D_{H_{2}O}=9.9\cdot 10^{-3}m^{2}s^{-1}\wedge D_{EtOH}=4.7\cdot 10^{-3}m^{2}s^{-1}$. This looks reasonable taking into account that we had only an approximation of the real asymptote of the permeation experiment.

Using the Equation (12) we can evaluate the diffusion coefficients of water and ethanol (in the membrane) which gives $2.80\times 10^{-11}{\;m}^{2}s^{-1}$ and $3.32\times 10^{-11}\;m^{2}s^{-1}$ , respectively. This result can be compared with the literature data. Qian et al. [38] studied the pervaporation process in water desalination by using chitosan mixed matrix membranes. Despite that they considered a different mixture (water/NaCl), the obtained values of water diffusion coefficient for pristine chitosan membrane ($8.90\times 10^{-11}÷11.30\times 10^{-11}{\;m}^{2}s^{-1}$) were comparable with reported in this paper. Mulder and Smoldres [39] also determined water diffusion coefficients for the pervaporative dehydration of ethanol using cellulose acetate membranes. Also in this case, the obtained results are comparable to ours ($D_{H_{2}O}=1.30\cdot 10^{-11}{\;m}^{2}s^{-1}$).

In addition to validate the method we have also generated numerical mass flow data for a membrane of thickness *L* = 40 μm, $D=10^{-11\;}m^{2}s^{-1}$ and pipe of length *l_p_* = 2 m, $D_{p}=6.25\times 10^{-4}{\;m}^{2}s^{-1}$ (by random walk method). The resulting time lags read $L_{a}^{1}$ = 0.96 s, $L_{a}=-47.5s\;$(see Appendix A). Application of our method gives $D=\frac{-L^{2}}{3\left(L_{a}-6.5La^{1}\right)}=0.99\cdot 10^{-11}\;m^{2}s^{-1}$. This value is almost identical to the value introduced into the simulation.

## 5. Conclusions

New method to estimate the diffusion coefficient based on a series solution of the permeation problem has been developed and evaluated. The model assumes that before measurement the analysed membranes are not empty. It also considers the effect of the tubing length between the permeation cell and the cold traps which causes a delay in the first moments of the mass flow in the measurement. Computational and analytical results were addressed to the experimental results obtained for chitosan membranes cross-linked with glutaraldehyde. The procedures do not involve much sophisticated mathematical methods.

## Figures and Tables

**Figure 1 polymers-11-00343-f001:**
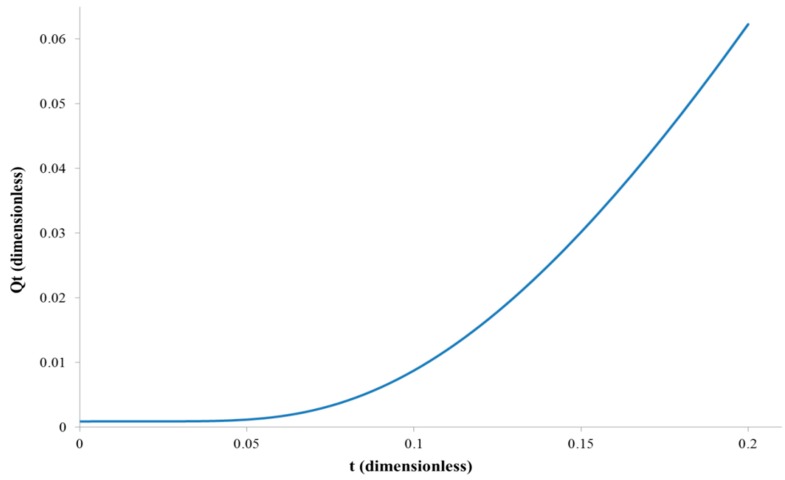
The relation between the total amount of diffusing substance and the time in the most common experimental permeation arrangement, i.e., C_0_ = C_2_ = 0.

**Figure 2 polymers-11-00343-f002:**
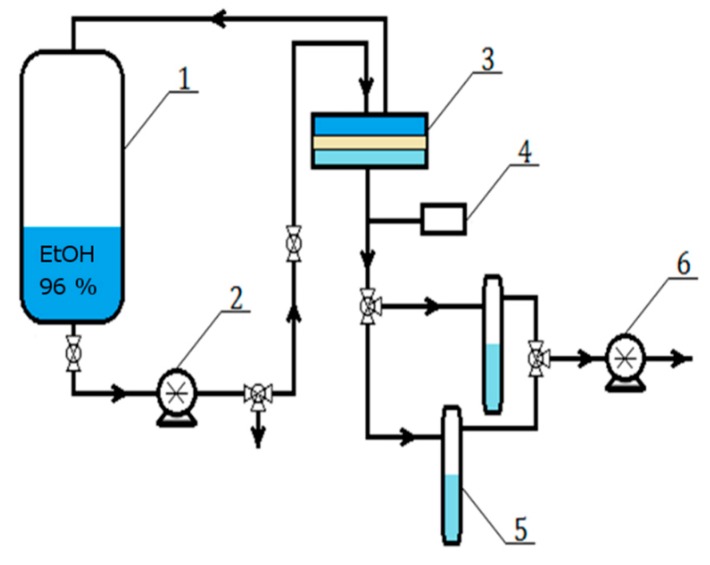
Scheme of pervaporation setup: 1—feed tank, 2—circulation pump, 3—separation chamber, 4—vacuum gauge, 5—cooled collection traps and 6—vacuum pump.

**Figure 3 polymers-11-00343-f003:**
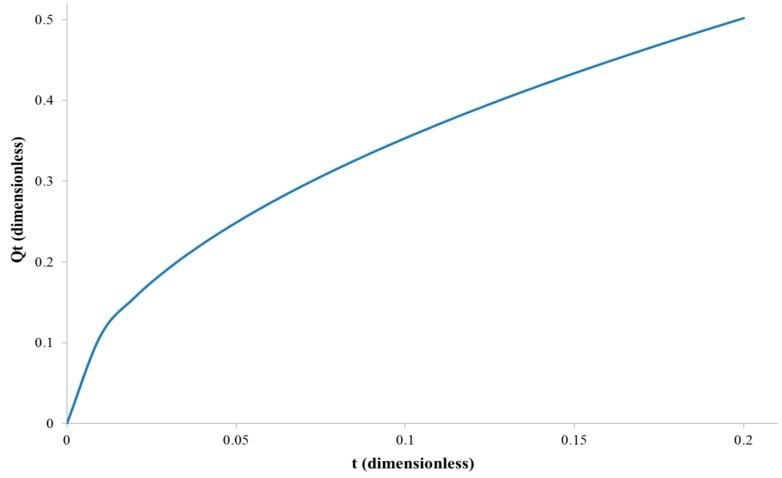
The relation between the total amount of diffusing substance and the time in the case when the membrane is filled with feed solution, i.e., *C*_0_ = *C*_1_.

**Figure 4 polymers-11-00343-f004:**
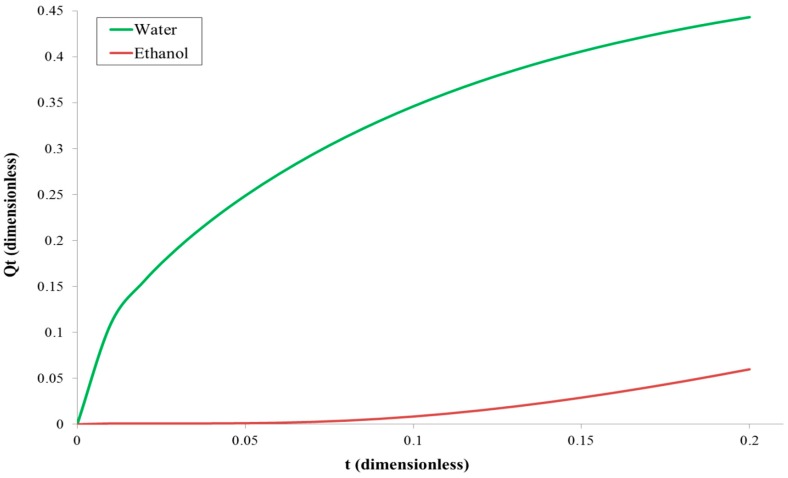
The relation between the total amount of diffusing water and ethanol molecules versus time in the case when the membrane is filled with water, i.e., *C*_0_ ≠ *C*_1_.

**Figure 5 polymers-11-00343-f005:**
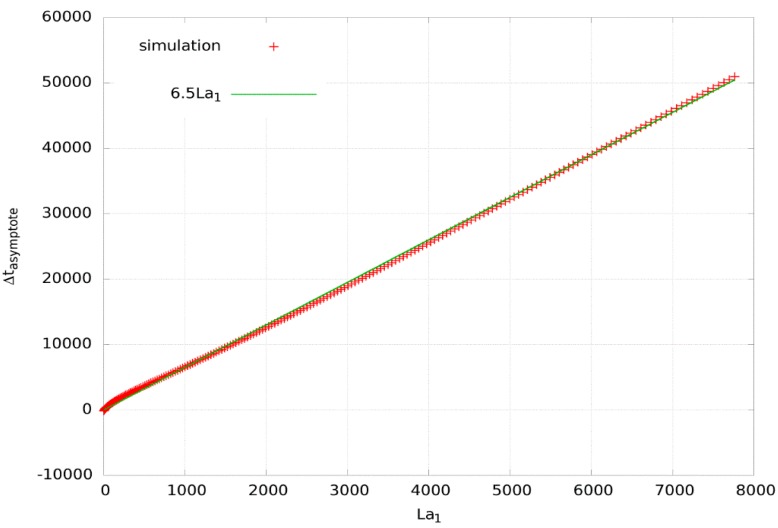
The experimental relation between $L_{a}^{1}$ and the delay of the mass permeation asymptote. Calculated numerically for different values of the diffusion coefficients and/or lengths of the tubing. The units of time are irrelevant for linear relation.

**Figure 6 polymers-11-00343-f006:**
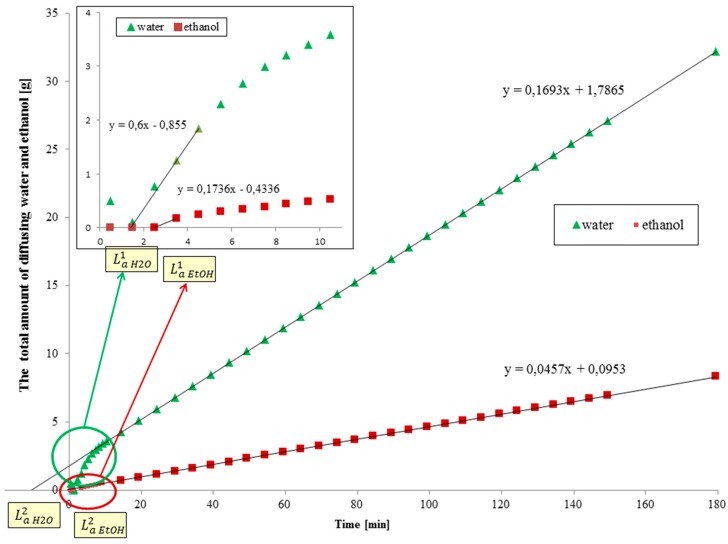
The total amount of diffusing water and ethanol separately versus time. The graphical way for calculating time lag for the short time $L_{a}^{1}$ and for the long-time $L_{a}^{2}$ for water and ethanol.

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
