# Peer review of "A Simple Methodology to Estimate the Diffusion Coefficient in Pervaporation-Based Purification Experiments"

_polymers, 2019, doi:10.3390/polym11020343_

Round 1

Reviewer 1 Report

The originality is moderate; it uses existing models to get diffusion coefficients in simple experiments. The paper is well written and easy to read. The conclusions are consistent with the evidence and arguments presented

Author Response

Answers for the Reviewer 1

 “The originality is moderate; it uses existing models to get diffusion coefficients in simple experiments. The paper is well written and easy to read. The conclusions are consistent with the evidence and arguments presented”

Answer:  We would like to thank the Reviewer for his opinion regarding our paper. We are glad us that our method of estimation of diffusion coefficient in pervaporation process is received as good and interesting.

Reviewer 2 Report

This manuscript describes a new method for obtaining the diffusion coefficient in pervaporation. The topic is interesting, but the manuscript is not well written and needs many corrections.

I suggest publication of this paper after major revision. Please take into consideration the following comments and suggestions in the preparation of the revised manuscript:

1.       The title does not reflect the content of the manuscript. Is this an easy way?

2.       Introduction

Line 40- Yeom et. al proposed not “found”. Line 50- determined not “found”.

Lines 74-75- Too much detail. Please move it to the experimental section.

Line 77- There are many references on studies of sorption and diffusion in pervaporation. Please add some references and replace “experimentalists” by researchers.

Line 90 – Replace “dissolution “  by sorption.

3.       Theoretical background

Lines 108 -110 – Units of variables are missing.

The values in the axis of Fig. 1 should have points not comas (0.1 not 0,1).

Line 131 -There are other methods for obtaining sorption coefficients!

Please replace formula by equation in the text.

4.       Experimental

Line 153 – Please replace “appliance” by set-up.

5.       Results and Discussion

Line 191 – C0=C1 only if the sorption coefficient is 1!

Line 220- It is volume fraction not concentration. Please check whether this affects the calculations.

Line 232- H is usually reported as Henry constant relating pressure and concentration of a solute in equilibrium between a gas and liquid phase. This is not the case!

Line 248- How were these simulations performed?

Line 311- The units of the diffusion coefficient should be m2/s. These values should be compared with other obtained by different methods. This would validate the new metod proposed.

Author Response

Answers for the Reviewer 2

Response to Reviewer 2 comment No 1:

“The title does not reflect the content of the manuscript. Is this an easy way?”

Answer: According to the Reviewer suggestion the title of presented manuscript was changed.

Response to Reviewer 2 comment No 2:

“Introduction

Line 40- Yeom et. al proposed not “found”. Line 50- determined not “found”.

Lines 74-75- Too much detail. Please move it to the experimental section.

Line 77- There are many references on studies of sorption and diffusion in pervaporation. Please add some references and replace “experimentalists” by researchers.

Line 90 – Replace “dissolution “  by sorption.”

Answer:  According to the Reviewer suggestion we changed “found” to “proposed” and “determined”, and “experimentalists” to “researchers”. Additionally we removed the details of pervaporation setup from Introduction. We added the references related to sorption and diffusion in pervaporation [27-30].

Response to Reviewer 2 comment No 3:

“3. Theoretical background

Lines 108 -110 – Units of variables are missing.

The values in the axis of Fig. 1 should have points not comas (0.1 not 0,1).

Line 131 -There are other methods for obtaining sorption coefficients!

Please replace formula by equation in the text.”

Answer: We agree with the Reviewer that in the description of parameters which appear in the equation (2) lack the units of variables. In such case of equation units depend on particular choise. Eg. Q [mol/m2], C [mol/m3], D [m2/s], t [s], l [m].

According to Reviewer suggestion we changed comas to points in the Figs. 1, 3, 4. Additionally we changed “formula” to “equation” in the whole manuscript.

We agree with the Reviewer that there are other methods for determining the sorption coefficient. We added the appropriate literature to our manuscript [27-30]. Despite of the fact that such methods characterized by simplicity in performance, they are not accurate for certain cases. For example, the solubility of pure ethanol in the polymer is extremely low, hence the measurement error is significant [28]. We added the information about sorption-desorption method for determination sorption coefficient in the Section “Theoretical background”.

Response to Reviewer 2 comment No 4:

“4. Experimental - Line 153 – Please replace “appliance” by set-up.”

Answer: We thanks the Reviewer for his correction. We changed “appliance” to “set-up.”

Response to Reviewer 2 comment No 5:

“5. Results and Discussion

Line 191 – C0=C1 only if the sorption coefficient is 1!

Line 220- It is volume fraction not concentration. Please check whether this affects the calculations.

Line 232- H is usually reported as Henry constant relating pressure and concentration of a solute in equilibrium between a gas and liquid phase. This is not the case!

Line 248- How were these simulations performed?

Line 311- The units of the diffusion coefficient should be m2/s. These values should be compared with other obtained by different methods. This would validate the new method proposed.”

Answer: We agree with the reviewer that if C0 is the feed concentration then C0=C1 only if the sorption coefficient is 1, but in our case we considered C0 in the membrane – not in the feed. In this case C0=C1.

We agree with the Reviewer that we considered volume fraction – not concentration. Fortunately, these two parameters are proportional, so this does not influence the calculation, where we consider a ratio of these fractions for single component.

According to the Reviewer suggestion we changed the symbol “H” that, of course, is usually reported as Henry constant, to “K”.

According to the Reviewer suggestion we added the information about the performed type of simulation in the Section Considering the effect of tubing between the membrane and the cold trap.

According to the Reviewer suggestion we changed the unit of diffusion coefficient from cm2/s to m2/s. We compared estimated values of diffusion coefficients with the result reported in the papers [39-40]. Although the citied works use different experimental conditions (comparable, but different membranes, different feed composition), the obtained values of water diffusion coefficient are the same order of magnitude as the values estimated in our experiment. In additional to validate the method we have also generated numerical mass flow data. The appropriate information was added to the manuscript (Section Application to Chitosan Pervaporation Membrane).

Round 2

Reviewer 2 Report

The authors addressed the comments suggested by the reviewer and modified the manuscript accordingly.

I suggest publication after minor revisions. I suggest the following:

Title:

A simple methodology to estimate the diffusion coefficient in  pervaporation based purification experiments

Line 110 Please specify the units.  

Author Response

Thank you for the review.

We have changed the title and added the missing units according to the suggestion (there must have been an editing error in the previous revision). We have also re-read the manuscript to fix language issues.